# Greater Risk of Pregnancy Complications for Female Surgeons: A Cross-Sectional Electronic Survey

**DOI:** 10.3390/ijerph20010125

**Published:** 2022-12-22

**Authors:** Fleur Delva, Pierre Carcasset, Pauline Mouton, Rivana Auguste-Virginie, Fanny Lairez, Loïc Sentilhes, Patrick Brochard, Jean-Philippe Joseph

**Affiliations:** 1Environmental Health Platform Dedicated to Reproduction, ARTEMIS Center, Bordeaux University Hospital, 33076 Bordeaux, France; 2Bordeaux Population Health Research Center, Inserm UMR1219, University of Bordeaux, 33076 Bordeaux, France; 3Centre d’Investigation Clinique 1401 Épidémiologie Clinique, Institut National de la Santé et de la Recherche, University of Bordeaux, 33076 Bordeaux, France; 4Department of General Practice, University of Bordeaux, 33076 Bordeaux, France; 5Department of Obstetrics and Gynecology, Bordeaux University Hospital, 33076 Bordeaux, France

**Keywords:** occupational exposure, pregnancy complications, female surgeons

## Abstract

Background: Female surgeons are exposed to physical and mental stressors that differ from those of other specialties. We aimed to assess whether female surgeons are more at risk of pregnancy complications than women in other medical specialties. Methods: We used a cross-sectional electronic survey of female physicians working in the French Region Nouvelle-Aquitaine who were pregnant between 2013 and 2018. A pregnancy complication was defined as the occurrence of miscarriage, ectopic pregnancy, fetal growth restriction, prematurity, fetal congenital malformation, stillbirth, or medical termination of the pregnancy. Multivariate logistic regression models were used to evaluate the risk of pregnancy complications for female surgeons relative to women practicing in other medical specialties. Results: Among the 270 women included, 52 (19.3%) experienced pregnancy complications and 28 (10.4%) were surgeons. In the multivariate analysis, female surgeons had a higher risk of pregnancy complications: adjusted odds ratio = 3.53, 95% confidence interval [1.27–9.84]. Conclusion: Further research is necessary to identify the hazards specifically involved in the practice of surgery to be able to propose preventive actions targeted to female surgeons during their pregnancy.

## 1. Introduction

Occupational exposure to certain risk factors is known to interfere with pregnancy. Chemical substances, such as organic solvents, have an adverse effect on pregnancy outcomes [1]. Biological agents that affect pregnancy are also well identified in the scientific literature [2]. They either affect the fetus directly (toxoplasmosis) or indirectly through the hyperthermia that results from infection (influenza). Ionizing radiation and biomechanical and organizational constraints have also been reported to interfere with pregnancy [3,4]. The profession of being a physician can expose women to such hazards. For example, exposure to antineoplastic agents can occur, either by direct contact or by contamination, and concerns a large number of health professionals, including physicians. Although no studies have been specifically conducted on physicians, one meta-analysis published in 2005 of 14 studies found a higher risk of spontaneous abortion following exposure to chemotherapy [5]. A prospective cohort study in the US conducted on nurses subjected to various conditions of exposure found antineoplastic drug exposure to be associated with a two-fold increased risk of spontaneous abortion [6]. In terms of anesthetic gases, very few studies have been conducted on the physician population. However, according to Warembourg et al., 2017 [7], studies conducted after the 2000s suggest a persistent increased risk of certain birth defects, but data are lacking concerning the risks of fetal death. There is also an excess risk of spontaneous abortion, preterm labor, and preterm delivery associated with night work [8,9]. In addition, despite the increase in the number of women in the medical profession, there is still a stigmatization attached to pregnant women, particularly in relation to the increased workload of colleagues [10]. This stigmatization can lead to a delay in pregnancy and to work later during pregnancy.

Although being a physician can expose women to such risks, the characteristics of exposure vary depending on the medical specialty. In 2021, an American study showed a greater risk of major pregnancy complications among female surgeons than among the wives of male surgeons [11], without specifying the professional activity of the surgeons’ wives. In 2017, another study found no difference in the proportion of women reporting missing work due to preterm labor between female physicians practicing in nonprocedural and procedural fields [12]. Occupational reproductive hazards identified in female physicians are radiation, blood-borne pathogens, infectious agents, anesthetic gases, toxic pharmaceuticals such as chemotherapeutic agents, surgical fumes, and the working conditions [10]. Female surgeons are exposed to physical and mental stressors that differ from those practicing in other specialties [10,13]. Surgical practice combines exposure to anesthetic gases, physical and organizational constraints (long working hours, shift work, prolonged periods of standing, and physically demanding work), psychological stress, and ionizing radiation. In animal studies, anesthetic gases are associated with reproductive disorders (teratogenic, developmental toxicity) [14]. Concerning physical and organizational constraints, the biological mechanisms involved in pregnancy complications are the alteration of placental perfusion and endocrine disruption [15,16]. In view of the occupational exposures of female surgeons, we hypothesize that female surgeons are more at risk of pregnancy complications than women in other medical specialties.

The objective of this study was therefore to assess whether female surgeons are more at risk of pregnancy complications than women in other medical specialties.

## 2. Materials and Methods

### 2.1. Study Cohort and Design

Our study consisted of a cross-sectional electronic survey of female physicians working in a French Region (Nouvelle-Aquitaine). Our study included women under the age of 50 who had been pregnant at least once between 2013 and the day of the survey. Women who were pregnant when the questionnaire was administered, who did not work during their last pregnancy, did not complete the entire survey, or were unemployed during their last pregnancy were excluded. We developed an anonymous self-administered survey composed of seven parts. The survey was developed to extract information on their sociodemographic characteristics, medical and obstetric history, and occupation. The survey was sent by e-mail to all female physicians practicing in Nouvelle-Aquitaine by the councils of the Order of Physicians of each department of Nouvelle-Aquitaine in December 2018. The e-mail contained a link to the survey, created using LimeSurvey^®^ tool (Hambourg, Germany), and a cover letter explaining the purpose of the study. A reminder was sent after 15 days.

### 2.2. Outcome

A pregnancy complication was defined as the occurrence of at least one of the following during pregnancy: miscarriage, extra-uterine pregnancy, fetal growth restriction, prematurity (birth before 37 weeks of amenorrhea), fetal congenital malformation, stillbirth, or medical termination of pregnancy. For each pregnancy complication, women were asked in the questionnaire if a medical diagnosis of each complication had been made during the pregnancy.

### 2.3. Exposure

The surgical specialties included the following: general surgery, gynecological and obstetrical surgery, maxillofacial surgery and stomatology, orthopedic and trauma surgery, plastic and cosmetic surgery, thoracic, cardiothoracic and vascular surgery, urological surgery, visceral and digestive surgery, ophthalmology, and otorhinolaryngology.

### 2.4. Statistical Analysis 

Quantitative analyses were carried out using SAS 9.4 software (Cary, NC, USA). The odds ratios (ORs) and confidence intervals (95% CIs) were estimated using multivariate logistic regression models. The analysis was performed on complete data (less than 5% missing data). We adjusted for potential confounders identified from the literature. Thus, potential confounders included in the multivariate logistic regression model were selected a priori from literature review: age ≥ 35 years at pregnancy, being primiparous, more than four physical and organizational constraints in the second or third trimester (being on call [night work or night and weekend shifts], home visits, prolonged standing [>6 h per day], working >42 h per week or carrying a heavy load [>15 kg]), infection by a teratogenic virus during pregnancy, exposure to at least one chemical agent (cytostatics, anesthetic gases, nitrous oxide), exposure to ionizing radiation, active smoking during pregnancy, alcohol consumption during pregnancy, the presence of a medical or surgical history (including uterine malformation, conization/myomectomy/scarred uterine surgery, cervical cancer, symptomatic/voluminous uterine fibroids, ovarian cyst, breast cancer, hypertension, type 1 diabetes, type 2 diabetes, thromboembolic disease, dysthyroidism, epilepsy, chronic inflammatory disease, autoimmune disease, depression, pelvic trauma, neoplasia), and the presence of an obstetrical history (recourse to assisted reproductive technology for the current pregnancy, stillbirth, voluntary termination of pregnancy, medical termination of pregnancy, prematurity, hypertensive disorders, cesarean section, premature rupture of membranes in the 2nd trimester, abruption placentae). The power of the study was calculated according to the number of female surgeons and women from other specialties, along with the proportion of pregnancy complications among women from other specialties, the OR found in the multivariate analysis, and an alpha risk of 0.05. With the data obtained, the power of the study was estimated to be 80%.

This study was approved by the institutional review board (ethics protection committee in November 2018) and conducted in compliance with MR-003 of the Commission Nationale de l’Informatique et des Libertés CNIL and conformed to General Data Protection Regulations.

## 3. Results

Among the 4788 emails sent by the council of the Order of Physicians, 995 (19.9%) surveys were completed. Among them, those of 270 (28.9%) women were included in the study (see flowchart—Figure 1).

Population characteristics by specialty are presented in Table 1. Surgeons experienced more physical or organizational constraints, exposure to chemical agents and exposure to ionizing radiation than physicians of other specialties (Table 1). There were three multiple pregnancies, two of which occurred among female surgeons. 

Among the 270 women included, 52 (19.3%) experienced pregnancy complications according to our definition. The most common pregnancy complication was fetal growth restriction N = 27 (10.0%), followed by miscarriage N = 16 (5.9%). No women had a stillbirth (Table 2).

In univariate analysis, female surgeons had a higher risk of pregnancy complications: odds ratio (OR) = 3.17 95% CI [1.38–7.27] (Table 3). The other factors (age, primiparous, organizational constraints, infection, exposure to chemical agent, exposure to ionizing radiation, smoking, alcohol, history) were not associated with pregnancy complication. 

In multivariate analysis, female surgeons had a higher risk of pregnancy complications aOR = 3.53, 95% CI [1.27–9.84]. The Hosmer and Lemeshow goodness-of-fit test did not reject the goodness-of-fit of the model (*p* = 0.56)

## 4. Discussion

### 4.1. Summary of the Study Findings 

This study suggests that female surgeons have a higher risk of pregnancy complications than women practicing other specialties. 

### 4.2. Comparison with Other Studies

Few studies have examined the effects of surgery on pregnancy. A study conducted in the United States that compared female surgeons to surgeons’ wives reported similar results in 2020 [11]. Similar results were also found in a study conducted solely among female urologists [17] and orthopedic surgeons working >60 h/week [18]. Another study failed to confirm these results, but compared procedural specialties, including anesthesiologists and gastroenterologists, to non-procedural specialties [12]. Thus, there appears to be an increased risk of pregnancy complications in female surgeons. In our study, we found a higher prevalence during pregnancy of organizational constraints (on-call duty, prolonged standing, work >42 h/week, heavy loads) during the second and third trimesters of pregnancy, of exposure to at least one chemical agent, to anesthetic gases, to nitrous oxide and to ionizing radiation in female surgeons than in other specialties. Thus, the cumulative exposure of female surgeons during pregnancy could explain this increased complication rate. In fact, in the general population, it has been shown that the accumulation of constraints leads to pregnancy complications [19,20].

In our sample, the complication rate for fetal growth restriction was similar to that of the French national perinatal survey, which was representative of the general French population [21]. However, the prematurity rate was much lower (2% vs. 11.5% in the perinatal survey). The miscarriage rate was also lower than that generally reported in the literature, which is approximately 20%. However, in the literature, female physicians have been reported to be more at risk of complications than the general population [10]. It is possible that women who had pregnancy complications responded less to our questionnaire.

### 4.3. Explanation of Findings

There were very few active smokers in this sample of female physicians relative to the frequency found in the general population [22]. The prevalence of active smokers during pregnancy has been shown to decrease with increasing income [23]. Moreover, female physicians have a medical knowledge of the effects of tobacco on pregnancy, which would encourage them to not smoke during pregnancy. Surprisingly, nearly 8% of the women drank alcohol during pregnancy, whereas prevention messages in France target zero alcohol consumption during pregnancy. As the data in the scientific literature on the effects of occasional low-dose alcohol consumption on pregnancy are less well established, it is possible that female doctors allowed themselves exceptional consumption of alcohol but not tobacco.

Among occupational risk factors, the best known and most studied is ionizing radiation. To date, it appears that the preventive measures in place are sufficient to avoid pregnancy complications. The International Commission on Radiological Protection recommends that a pregnant woman be exposed to no more than 1 millisievert during pregnancy. Data from the literature show teratogenic effects for exposure to approximately 200 mGy; however occupational exposures does not generally reach this level [24]. In a review conducted in 2020, no studies were found that reported exposure above this recommended level using current protective measures [25]. In 2016, a study conducted among interventional radiologists reported no difference in terms of an effect on fetal health relative to the general population [26]. The effect of anesthetic gases could be related to inhibitory effects on cell division, resulting in an increase in the number of abnormal cells and chromosomal aberrations [27]. A review of the literature suggests an increased risk of congenital malformations, particularly if gas exposure is not controlled and persists above the recommended limits [7]. Exposure limits may be exceeded in computed tomography scan/magnetic resonance imaging units, ambulatory operating rooms, burn units, otorhinolaryngology units, and pediatric surgery units [28]. Factors that could be involved in exceeding the recommended limits include interventions on children or patients for whom face mask-based ventilation systems are used [28]. The possible effects of surgical gases on reproduction have not been reported in the scientific literature. However, as stated in a review of the literature published in 2020, several compounds in surgical gases have been shown to have an effect on pregnancy and child development, such as fine particles, benzene, toluene, etc. [25]. The main sources of ultrafine particles include combustion and exposure to ultrafine carbon particles (mainly from combustion) in the workplace and have been shown to be associated with a risk of low birth weight [29]. Animal studies have shown that ultrafine particles can act by a direct mechanism of toxicity on the placenta, as well as by indirect mechanisms through oxidative stress and inflammation [30]. Finally, surgeons are subject to physical and organizational constraints. Working ≥40 h per week, night work, prolonged standing, carrying heavy loads, and the accumulation of constraints have been shown to be associated with preterm birth, low birth weight, and spontaneous miscarriage [3,6,8,9]. 

Other factors that may be involved in the observed increased risk of pregnancy complications include the length of training for surgeons, especially post-internship training, which often leads to a later age of maternity than in other medical specialties [12]. 

### 4.4. Strengths and Limitations

In our study, we were unable to investigate which risk factors are predominantly involved in the excess risk of complications because of the small sample size. In addition, the surgical specialties included a range of specialties that are heterogeneous in terms of exposure. It is therefore possible that the risk of pregnancy complications was overestimated for certain surgical specialties and underestimated for others.

Our study was questionnaire based, with the expected 20% response rate for this type of study. Such a low response rate could have created a selection bias but the distribution of responding specialties was similar to the regional distribution [31]. Another limitation of this type of retrospective questionnaire is the self-reporting of pregnancy complications. However, because our sample was composed solely of female physicians, there should have been little misclassification of the reported diagnoses.

### 4.5. Implications for Healthcare Practice

Female surgeons are exposed to a number of recognized risk factors for reproduction and although there are preventive measures in place, there appears to be an excess risk of pregnancy complications. It is possible that preventive measures are not optimally followed by female surgeons during pregnancy or are implemented too late once the pregnancy is known and declared. In France, pregnant women are not obliged to declare their pregnancy to their employer. However to benefit from pregnancy-related leave, the employer must be informed. For salaried women, maternity leave for a first child is six weeks before the presumed date of delivery and 10 weeks after that date. For self-employed women, to receive benefits, they must take a minimum of eight weeks maternity leave, six of which must be taken after the birth. Moreover, concerning physical and organizational constraints, there are few preventive measures that can be put in place other than adapting the workstation in terms of time spent working and posture. In a 2016 study in Germany, 62% of female surgeons continued to operate during their pregnancy [32]. The main reasons given were the joy of surgery and the team spirit. Nonetheless, 15% of the women said their department head expected them to continue working. Indeed, discrimination against women in surgery is still prevalent, in particular concerning their becoming pregnant, which can limit the implementation of preventive measures [33].

## 5. Conclusions

In conclusion, we have shown that female surgeons are at greater risk of pregnancy complications than women in other medical specialties. Indeed, female surgeons are exposed to a certain number of hazards that are associated in the scientific literature with an excess risk of pregnancy complications. Further studies are necessary to identify the hazards specifically involved in the practice of surgery to be able to propose targeted preventive actions to female surgeons during their pregnancy.

## Figures and Tables

**Figure 1 ijerph-20-00125-f001:**
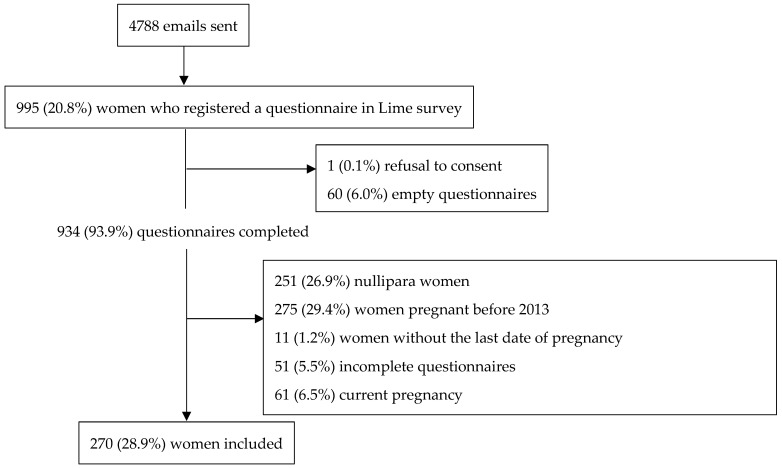
Flow chart of a cross-sectional electronic survey of female physicians working in a French region.

**Table 1 ijerph-20-00125-t001:** Main characteristics of the study population according to the specialties.

	Other Specialties N = 242 N (%)	Surgeon N = 28 N (%)	*p*-Value
Age at last pregnancy Mean (Standard Deviation)	34.5 (4.0)	33.8 (3.5)	0.27
Age ≥ 35 years	99 (40.9)	13 (46.4)	0.57
Primiparous	63 (26.0)	7 (25.0)	0.90
Obstetrical history *	61 (25.2)	10 (35.7)	0.23
Surgical or medical history **	70 (28.9)	6 (21.4)	0.40
Active smoking	13 (5.4)	2 (7.1)	0.70
Alcohol	18 (7.4)	3 (10.7)	0.54
Four or more physical or organizational constraints	10 (4.1)	4 (14.3)	0.04
On call duty in the 2nd and 3rd trimesters of pregnancy	69 (28.5)	17 (60.7)	0.0005
Home visits in the 2nd and 3rd trimesters of pregnancy	91 (37.6)	0 (0.0)	<0.0001
Prolonged standing in the 2 and 3rd trimesters of pregnancy	84 (34.7)	17 (60.7)	0.007
Work > 42 h/week in the 2nd and 3rd trimesters of pregnancy	113 (46.7)	19 (67.9)	0.04
Heavy loads in the 2nd and 3rd trimesters of pregnancy	14 (5.8)	5 (17.9)	0.03
Teratogen infection during pregnancy ***	4 (1.6)	1 (3.6)	0.47
Exposure to at least one chemical agent	50 (20.7)	23 (82.1)	<0.0001
Exposure to a cytostatic	8 (3.3)	2 (7.1)	0.31
Exposure to anesthetic gases	10 (4.1)	19 (67.9)	<0.0001
Exposure to nitrous oxide	43 (17.8)	17 (60.7)	<0.0001
Exposure to ionizing radiation	21 (5.4)	12 (42.9)	<0.0001

* Recourse to assisted reproductive technology for the current pregnancy, stillbirth, voluntary termination of pregnancy, medical termination of pregnancy, prematurity, hypertensive disorders, cesarean section, premature rupture of membranes in the 2nd trimester, abruption placentae; ** Uterine malformation, conization/myomectomy/scarred uterine surgery, cervical cancer, symptomatic/voluminous uterine fibroids, ovarian cyst, breast cancer, hypertension, type 1 diabetes, type 2 diabetes, thromboembolic disease, dysthyroidism, epilepsy, chronic inflammatory disease, autoimmune disease, depression, pelvic trauma, neoplasia), or the presence of an obstetrical history; *** rubella, cytomegalovirus, varicella-zoster virus, parvovirus B19, toxoplasmosis, listeria, Zika.

**Table 2 ijerph-20-00125-t002:** Pregnancy complications among 270 female physicians.

	Total Population	Female Surgeons	Other Specialities	
Pregnancy Complication	N (%)	N (%)	N (%)	*p*-Value
Miscarriage	16 (5.9)	1 (3.6)	15 (6.2)	0.5
Extra-uterine pregnancy	4 (1.5)	0 (0.0)	4 (1.6)	1.0
Fetal growth restriction	27 (10.0)	8 (28.6)	19 (7.8)	0.003
Prematurity	6 (2.2)	4 (14.3)	2 (0.8)	0.001
Fetal congenital malformation	2 (0.7)	0 (0.0)	2 (0.8)	1.0
Stillbirth	0 (0.0)	0 (0.0)	0 (0.0)	-
Medical termination of pregnancy	1 (0.4)	0 (0.0)	1 (0.4)	1.0

The same woman could have had several complications.

**Table 3 ijerph-20-00125-t003:** Risk of pregnancy complications for female surgeons relative to women in other medical specialties—univariate logistic regression model.

	OR	95% CI	*p*
Surgery specialty (ref other specialties)	3.17	[1.38–7.27]	6
Age ≥ 35 years	1.27	[0.69–2.33]	0.44
Primiparous	0.83	[0.41–1.69]	0.60
Four or more physical or organizational constraints	2.47	[0.79–7.71]	0.12
Teratogen infection during pregnancy	2.87	[0.47–17.6]	0.25
Exposure to at least one chemical agent	1.25	[0.64–2.43]	0.50
Exposure to ionizing radiation	0.77	[0.41–1.46]	0.42
Active smoking	1.57	[0.48–5.14]	0.45
Alcohol	0.98	[0.32–3.06]	0.98
Surgical or medical history	1.04	[0.53–2.04]	0.90
Obstetrical history	1.84	[0.97–3.50]	0.06

OR: odds ratio, 95% CI: 95% confidence interval.

## Data Availability

The datasets used during the current study are available from the corresponding author on reasonable request.

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
