# Peer review of "Greater Risk of Pregnancy Complications for Female Surgeons: A Cross-Sectional Electronic Survey"

_ijerph, 2022, doi:10.3390/ijerph20010125_

Round 1
Reviewer 1 Report
In the section Introduction, the authors should clearly explain the research problem and the objectives of the study. The authors need to articulate the ‘missing puzzle piece’ their research aims to cover more clearly and specifically. I usually recommend the logical flow for the introduction to be: 1) what is the problem and why is it important, 2) what we know, 3) what we don’t know, and finally, 4) what are we doing about it.
Authors should formulate hypotheses. And then, based on the results, confirm or reject the hypotheses.
The section Discussion provides a simple summary of the outcomes, that support all the hypotheses. This however raises the question, what is new about the outcome of this research? This needs to be emphasized more clearly when putting together the literature review and the conceptual framework and discussed in the discussion/results section to clarify the contribution of the paper to the body of knowledge.
Author Response
We thank the referee for these valuable comments and suggestions. We made changes in the article according to your advices to make it clearer and more exhaustive. The modifications are described hereafter and we give a point-by-point response to the concerns.
In the section Introduction, the authors should clearly explain the research problem and the objectives of the study. The authors need to articulate the ‘missing puzzle piece’ their research aims to cover more clearly and specifically. I usually recommend the logical flow for the introduction to be: 1) what is the problem and why is it important, 2) what we know, 3) what we don’t know, and finally, 4) what are we doing about it.
We have modified the order to the introduction
Authors should formulate hypotheses. And then, based on the results, confirm or reject the hypotheses.
We have added hypothesis
The section Discussion provides a simple summary of the outcomes, that support all the hypotheses. This however raises the question, what is new about the outcome of this research? This needs to be emphasized more clearly when putting together the literature review and the conceptual framework and discussed in the discussion/results section to clarify the contribution of the paper to the body of knowledge.
We have added in the discussion data to explain our results

Reviewer 2 Report
This study investigated whether female surgeons are at a greater risk for a complicated pregnancy. The authors found that female surgeons experienced more physical/organizational constraints, exposure to chemical agents, and exposure to ionizing radiation than other medical specialties.
Importantly, female surgeons were more likely to experience a pregnancy complication, irrespective of known confounders. Overall, the manuscript is well written. I have a few comments below:
Abstract:
Page 1, Line 18 - What type of hypotrophy are the authors referring to?
Introduction:
Page 2, Line 46 – Should ‘excess’ be ‘increased’?
Methods:
2.6 Statistics - Did the authors look for other pregnancy complications apart from fetal growth restriction, such as preeclampsia or gestational diabetes?
Results:
Table 2 – It would be useful to include the incidence of pregnancy complications in the ‘Other Specialty’ category of physicians. As well as the statistical comparison between physician categories.
Did the authors stratify the data to see whether female surgeons are at a greater risk of specific pregnancy complications than other speciality physicians?
Discussion:
It is possible that the increased physical and organizational constraints in female surgeons causes increased stress, which can impair fetal growth. Do the authors have any indication of stress/anxiety in the women from their questionnaire?
Author Response
We thank the referee for these valuable comments and suggestions. We made changes in the article according to your advices to make it clearer and more exhaustive. The modifications are described hereafter and we give a point-by-point response to the concerns
Abstract:
Page 1, Line 18 - What type of hypotrophy are the authors referring to?
This is an error, we have delete this term
Introduction:
Page 2, Line 46 – Should ‘excess’ be ‘increased’?
We have modified this term
Methods:
2.6 Statistics - Did the authors look for other pregnancy complications apart from fetal growth restriction, such as preeclampsia or gestational diabetes?
No in this study we didn’t look preeclampsia and gestational diabetes. We have only looked miscarriage, extra-uterine pregnancy, fetal growth restriction, prematurity (birth before 37 weeks of amenorrhea), fetal congenital malformation, stillbirth, or medical termination of pregnancy.
Results:
Table 2 – It would be useful to include the incidence of pregnancy complications in the ‘Other Specialty’ category of physicians. As well as the statistical comparison between physician categories.
We have added these data
Did the authors stratify the data to see whether female surgeons are at a greater risk of specific pregnancy complications than other speciality physicians?
In view of the small number of subjects, we have chosen not to carry out analyses by type of complication because this would lead to a lack of power
Discussion:
It is possible that the increased physical and organizational constraints in female surgeons causes increased stress, which can impair fetal growth. Do the authors have any indication of stress/anxiety in the women from their questionnaire?
Indeed, but we did not ask them because it seemed complicated in a retrospective survey to assess stress and anxiety during the past pregnancy

Reviewer 3 Report
it is a very interesting article, with important relevance into the clinical practice.
I would have some suggestions/questions:
1. in the introduction there should be more information about the mental stressors that come with that - the necessity to work despite being pregnant, the discrimination- should be more elaborate in the introduction, as we can find them in the conclusion
2. patients age could also be a cofounder , a subgroup analysis
3. did they have problems with previous pregnancies
Author Response
We thank the referee for these valuable comments and suggestions. We made changes in the article according to your advices to make it clearer and more exhaustive. The modifications are described hereafter and we give a point-by-point response to the concerns.
- in the introduction there should be more information about the mental stressors that come with that - the necessity to work despite being pregnant, the discrimination- should be more elaborate in the introduction, as we can find them in the conclusion
We have added this information
- patients age could also be a cofounder , a subgroup analysis
Yes, we think that age is a cofounder, so we have adjusted on this factor in multivariable analysis
- did they have problems with previous pregnancies
Yes, this data is in obstetrical history. And we have adjusted also on obstetrical history
